# Equality Constraints in Linear Hawkes Processes

**Søren Wengel Mogensen**                    SOREN.WENGEL_MOGENSEN@CONTROL.LTH.SE
*Department of Automatic Control, Lund University, Sweden*

**Editors:** Bernhard Schölkopf, Caroline Uhler and Kun Zhang

## Abstract

Conditional independence is often used as a testable implication of causal models of random variables. In addition, equality constraints have been proposed to distinguish between data-generating mechanisms. We show that one can also find equality constraints in linear Hawkes processes, extending this theory to a class of continuous-time stochastic processes. This is done by proving that Hawkes process models in a certain sense satisfy the equality constraints of linear structural equation models. These results allow more refined constraint-based structure learning in this class of processes. Arguing the existence of equality constraints leads us to new identification results for Hawkes processes. We also describe a causal interpretation of the linear Hawkes process which is closely related to its so-called cluster representation.

**Keywords:** equality constraints, causal identification, linear Hawkes processes, structure learning

## 1. Introduction

In causal inference, the question of what can be learned about the underlying structure from observational data is a classical one (Spirtes et al., 2000; Pearl, 2009). While most work studies models of random variables and without an explicit notion of time, this question has also been studied for stochastic process models. For a multivariate stochastic process, $X_t = (X_t^1, X_t^2, \ldots, X_t^n)^T$, this is often formalized by assuming that the dynamical evolution of each coordinate process, $X_t^i$, depends directly only on a subset of the other coordinate processes which can be represented by a directed graph which is the *structure* that methods aim to learn. As examples of this, Eichler et al. (2017); Xu et al. (2016) learn graphs describing the structure of multivariate linear Hawkes processes from dynamical observation of the process whereas Achab et al. (2017) use integrated cumulants of linear Hawkes processes for structure learning. These approaches assume that we have full observation in the sense that every coordinate process of the system is observed. Meek (2014); Mogensen et al. (2018); Mogensen (2020) consider structure learning based on tests of so-called local independence and provide methods that can recover structural information based on a marginal distribution only.

In this paper, we show that one can use equality constraints in linear Hawkes processes to distinguish between graphical structures that are indistinguishable using tests of local independence only which enables more refined structure learning algorithms. This development parallels that of structural causal models without an explicit notion of time in which equality constraints have been used for structure learning in addition to conditional independence constraints (Robins, 1986; Verma and Pearl, 1990; Robins, 1999; Tian and Pearl, 2002; Shpitser et al., 2014; Richardson et al., 2017; Bhattacharya et al., 2021; Wang and Drton, 2020).

Some results are not proven in the main text. Their proofs can be found in Appendix B.

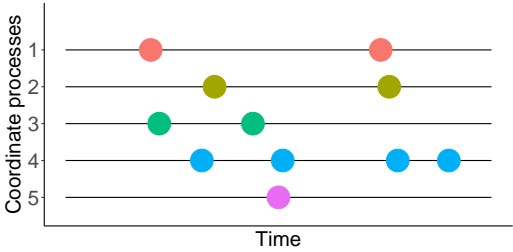 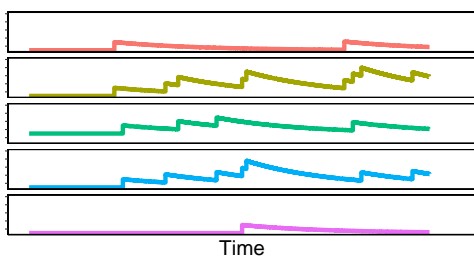

Figure 1: Left: Example of observation from 5-dimensional Hawkes process, $V = \{1, 2, 3, 4, 5\}$. Colors (and vertical placement) indicate coordinate process. For each event (dot) time of occurrence is known as well as which coordinate process the event occurs in. Right: intensity processes, $\lambda_t^\alpha, \alpha \in V$, corresponding to event histories from the left plot (for simplicity only displayed events count in the intensities). The intensities are stochastic processes and from the stochastic integral in Definition 1, one can see that an $\alpha$-event increases the $\beta$-intensity if $\phi_{\beta\alpha} \neq 0$.

## 2. Hawkes Processes

Hawkes processes have received much attention in machine learning in recent years as a tractable model of mutually exciting streams of events (Zhou et al., 2013; Luo et al., 2015; Etesami et al., 2016; Tan et al., 2018; Xu et al., 2018; Trouleau et al., 2019). The following is a brief introduction to this model class, see Laub et al. (2015), Liniger (2009), and Daley and Vere-Jones (2003) for additional background on Hawkes processes and point processes in general.

We consider a filtered probability space $(\Omega, \mathcal{F}, (\mathcal{F}_t), P)$ and an $n$-dimensional point process, $X_t = (X_t^1, X_t^2, \ldots, X_t^n)^T$, and let $V = \{1, 2 \ldots, n\}$. $\mathcal{F}_t$ is a filtration, that is, a nondecreasing family of $\sigma$-algebras which for each time point, $t \in \mathbb{R}$, represent the available information at that time. For each $\alpha \in V$, we say that $X^\alpha$ is a *coordinate process*. Each coordinate process is described by a series of *events*, or *points*, that occur at time points indexed by $\mathbb{R}$. For $\alpha \in V$, this is formalized by random variables, $\{T_i^\alpha\}_{i \in \mathbb{Z}}$, such that $T_i^\alpha < T_{i+1}^\alpha$ almost surely. For $s < t$, we let the $N_t^\alpha - N_s^\alpha$ denote the number of points (events) in the $\alpha$-process in the interval $(s, t]$, that is, $N_t^\alpha - N_s^\alpha = \sum_i \mathbb{1}_{s < T_i^\alpha \leq t}$. We assume that no two events occur simultaneously. For $\alpha \in V$, the *intensity process*, $\lambda_t^\alpha$, satisfies

$$\lambda_t^\alpha = \lim_{h \downarrow 0} \frac{1}{h} P(N_{t+h}^\alpha - N_t^\alpha = 1 \mid \mathcal{F}_t).$$

The intensity, $\lambda_t^\alpha$, therefore describes how likely an $\alpha$-event is in the immediate future $(t, t + h]$, for small $h$, given the past until time $t$.

**Definition 1 (Linear Hawkes process)** *We say that a point process, $X$, is a* linear Hawkes process *if for all $\beta \in V$ and $t \in \mathbb{R}$ the intensity is of the form*

$$\lambda_t^\beta = \mu_\beta + \sum_{\alpha \in V} \int_{-\infty}^t \phi_{\beta\alpha}(t - s) dN_s^\alpha$$

*for a constant $\mu_\beta \geq 0$ and nonnegative functions $\phi_{\beta\alpha}$ for $\alpha \in V$.*

We will often write *Hawkes process* instead of *linear Hawkes process*. We assume throughout for all $\alpha$ and $\beta$ that $\phi_{\beta\alpha}$ is a continuous function on $(0, \infty)$ and zero outside this interval. We define the $n \times n$ matrix $\Phi$ such that $\Phi_{\beta\alpha} = \int_{-\infty}^{\infty} \phi_{\beta\alpha}(s)\mathrm{d}s$. For a square matrix, $A$, its spectral radius, $\rho(A)$, is the largest absolute value of its eigenvalues, $\lambda_1, \ldots, \lambda_m$, i.e., $\rho(A) = \max_{i=1,\ldots,m}\{|\lambda_i|\}$. If $\rho(A) < 1$ then we say that $A$ is *Schur stable*. We assume that $\Phi$ is Schur stable and therefore we can assume the Hawkes process to have stationary increments and $\lambda_t$ to be a stationary process (Jovanović et al., 2015; Bacry and Muzy, 2016). We define the $n \times n$ matrix $R = (I_n - \Phi)^{-1}$ where $I_n$ is the $n \times n$ identity matrix. $R$ is well-defined due to the assumption on the spectral radius of $\Phi$ and furthermore $R = \sum_{i=0}^{\infty} \Phi^i$. For a matrix, $M$, with rows indexed by $I$ and columns indexed by $J$, we let $M_{\bar{I}\bar{J}}$ denote the submatrix indexed by $\bar{I} \subseteq I$ and $\bar{J} \subseteq J$. We let $dN_t^\alpha$ denote $N_{t+dt}^\alpha - N_t^\alpha$ where $dt$ is the differential of $t$. Following Hawkes (1971), for each $\alpha \in V$ we define

$$\Lambda^\alpha = E(dN_t^\alpha)/dt = E(\lambda_t^\alpha)$$

which is independent of $t$ due to stationarity. We let $\Lambda$ denote the $n \times n$ diagonal matrix such that $\Lambda_{\alpha\alpha} = \Lambda^\alpha$ and we assume $\Lambda$ to be positive definite.

**Integrated Covariance**   We define a probabilistic concept which is seen to be a cumulative measure of covariance between two coordinate processes. Let $\Sigma$ be an $n \times n$ matrix such that for $\alpha, \beta \in V$,

$$\Sigma_{\beta\alpha}dt = \int_{-\infty}^{\infty} E(dN_t^\beta dN_{t+\tau}^\alpha) - E(dN_t^\beta)E(dN_{t+\tau}^\alpha)d\tau. \tag{1}$$

The entries of $\Sigma$ contain the *integrated covariances* between pairs of coordinate processes. The following matrix equations hold (Jovanović et al., 2015; Achab et al., 2017),

$$\Sigma = R\Lambda R^T = (I_n - \Phi)^{-1}\Lambda(I_n - \Phi)^{-T}, \tag{2}$$

where $-T$ denotes transposition and inversion of a matrix.

**Cluster Representation**   In Definition 1, Hawkes processes are introduced by specifying their intensity processes. One can also define them using a so-called *cluster representation* (Jovanović et al., 2015). This representation lends itself to a straightforward causal interpretation as we will see.

The following describes the cluster representation which can be thought of as a simulation algorithm and it defines a Hawkes process given a set of constants $\{\mu_\alpha\}_{\alpha \in V}$ and a set of nonnegative functions $\{\phi_{\beta\alpha}\}_{\alpha, \beta \in V}$. For each $\alpha \in V$, *generation 0-events* are generated from a homogeneous Poisson process with rate $\mu_\alpha$. Each generation 0-event creates a *Hawkes tree*, that is, a cluster of events (the clusters are independent). The events in each cluster are generated recursively. Each event of type $\alpha$ at time $s$ in a cluster has *child* events of type $\beta$ according to an inhomogeneous Poisson process with rate $\phi_{\beta\alpha}(t - s)$ for $t > s$. This is illustrated in Figure 2 in which four generation 0-events are shown, each of them starting a cluster. Parent-child relations are shown using line segments. Note that $\phi_{\beta\alpha} = 0$ means that there will be no $\beta$-children from an $\alpha$-event, though there could be $\beta$-descendants further down from the $\alpha$-event, e.g., if there exists $\gamma$ such that $\phi_{\gamma\alpha} \neq 0$ and $\phi_{\beta\gamma} \neq 0$. The observed Hawkes process is then the superposition of the events in all

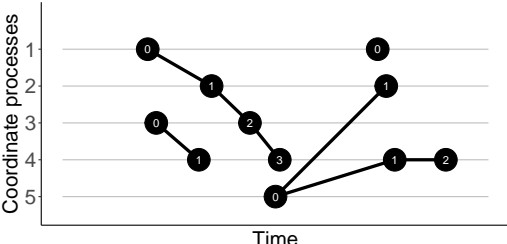

Figure 2: Same stream of events as in Figure 1 (left) showing the (unobserved) clusters of the events. Numbers indicate generation of the event and line segments indicate ancestry. For instance, the 1-2-3-4 cluster was produced from a generation 0-event of the 1-type which produced an event of the 2-type and so forth.

clusters. One should note that when observing a Hawkes process, we only have access to a set of event times and types $(\alpha_i, t_i)$ and not to the cluster structure, that is, the line segments in Figure 2 are unobserved.

The parameters in $\Phi$ and $R$ have straightforward interpretations. From the cluster process definition of the Hawkes process, it follows that $\Phi_{\beta\alpha}$ is the expected number of child events of type $\beta$ from an $\alpha$-event. More generally, for $i = 0, 1, 2, \ldots$, $(\Phi^i)_{\beta\alpha}$ is the expected number of $\beta$-descendants in the $i$'th generation of a Hawkes tree rooted at an $\alpha$-event. This implies that $R_{\beta\alpha}$ is the expected total number of $\beta$-descendants on such a tree (Jovanović et al., 2015).

### 2.1. Causal Interpretation

In this section, we will define what we mean by a *causal* Hawkes process. Mogensen (2020) defined a causal Hawkes process by requiring that the set of functions $\{\phi_{\beta\alpha}\}$ also describes how events enter the intensities in intervened systems where the trajectories of some coordinate processes are determined exogenously. We will instead use the cluster representation to give a causal interpretation based on a type of *injection intervention*. That is, the observational state of the system is given by the Hawkes process as defined above. An interventional state is defined by a set of deterministic *injections*, $\{(\alpha_i, t_i)\}_{i=1,\ldots,M}$, such that each injection is an element in $V \times \mathbb{R}$. An injection is a root in a Hawkes tree and the interventional processes are given as the superposition of all intrinsic and interventional trees. We then say that a linear Hawkes process is *causal* if a Hawkes tree rooted at an injection event has the same distribution as an intrinsic, or non-injection, Hawkes tree for all coordinate processes $\alpha \in V$ and all trees (interventional and intrinsic) are independent. In other words, the interventional events propagate in the same way through the system as the intrinsic events. This causal assumption is seen to correspond closely to the assumption of *autonomy* or *modularity* which is often used to define causal models (Pearl, 2009; Peters et al., 2017).

We note that this type of interventions is a simplistic version of interventions that are used in some fields of science, e.g., in neuroscience where experimenters may intervene on neurons by means of extracellular electrical stimulation. This, however, is not expected to break the intrinsic dynamics of the nervous system nor sever causal mechanisms (Meffin et al., 2012; Komarov et al., 2019). It is also similar to how interventions are often modelled in linear control theory in which they are external forces (possibly dependent on the past of the process) that change the dynamics of the system without necessarily breaking any connections (Åström and Murray, 2008; Zabczyk, 2020).

**Definition 2 (Causal effects)** *We will say that* $\Phi_{\beta\alpha}$ *is the* direct (causal) effect *from* $\alpha$ *on* $\beta$ *and that* $R_{\beta\alpha}$ *is the* total (causal) effect *from* $\alpha$ *on* $\beta$. *We will say that* $\phi_{\beta\alpha}$ *is the* (causal) link function *from* $\alpha$ *to* $\beta$.

The interpretation of the entries in $\Phi$ and $R$ and the causal assumption justify the above definitions. If we inject an interventional event of type $\alpha$ at time $s \in \mathbb{R}$, this event on average has $\Phi_{\beta\alpha}$ events of type $\beta$ as first-generation (or *direct*) descendants (when considering all $t > s$). The Hawkes tree rooted at the interventional $\alpha$-event on average has a total of $R_{\beta\alpha}$ events of type $\beta$, counting the injectional event if $\alpha = \beta$.

## 2.2. Graphical Representation

A *graph* is a pair $(V, E)$ where $V$ is a finite set of nodes and $E$ is a set of edges, each between a pair of (not necessarily distinct) nodes. We will use *directed mixed graphs* (DMGs), i.e., graphs such that every edge is either *directed*, $\rightarrow$, or *bidirected*, $\leftrightarrow$, and such that between any pair of nodes, $\alpha$ and $\beta$, there is a subset of the edges $\{\alpha \rightarrow \beta, \alpha \leftarrow \beta, \alpha \leftrightarrow \beta\}$. We use $\sim$ to denote a generic edge of either type. If an edge $\alpha \sim \beta$ is in a graph $\mathcal{G}$, we write $\alpha \sim_{\mathcal{G}} \beta$. If the edge $\alpha \rightarrow \beta$ is in the graph $\mathcal{G}$, we say that $\alpha$ is a *parent* of $\beta$, and we let $\text{pa}_{\mathcal{G}}(\beta)$ denote the set of parents of $\beta$ in the graph $\mathcal{G}$.

Given a causal Hawkes process with coordinate processes indexed by $V = \{1, 2, \ldots, n\}$, we construct its *causal graph*, $\mathcal{D} = (V, E)$, such that $\alpha \rightarrow_{\mathcal{D}} \beta$ if and only if $\phi_{\beta\alpha} \neq 0$. Using the continuity assumption of $\phi_{\beta\alpha}$, we see that if $\alpha \rightarrow \beta$ is in the causal graph, then there exists a nontrivial interval such that $\phi_{\beta\alpha} > 0$ on this interval.

A *walk* between $\alpha$ and $\beta$ is an ordered, alternating sequence of nodes, $\gamma_i$, and edges, $\sim_j$,

$$\alpha = \gamma_0 \sim_1 \gamma_1 \sim_2 \ldots \gamma_{m-1} \sim_m \gamma_m = \beta$$

such that for each $i = 1, \ldots, m$ the edge $\sim_i$ is between $\gamma_{i-1}$ and $\gamma_i$. We say that $\alpha$ and $\beta$ are *endpoint nodes*. A *path* is a walk such that no node occurs more than once. We say that a path from $\alpha$ to $\beta$ is *directed* if every edge is directed and points towards $\beta$, and a *directed cycle* is a directed path from $\alpha$ to $\beta$ composed with the edge $\beta \rightarrow \alpha$. We say that a DMG is *acyclic* if it contains no directed cycles. We say that a DMG is a *directed graph* if it only contains directed edges.

## 3. Representation as a Linear SEM

The relation between the observed quantity, $\Sigma$, and the parameters, $(\Phi, \Lambda)$, in Equation (2) is seen to be similar to the relation between the parameters and the covariance matrix of a *linear structural equation model* (linear SEM) and we will leverage this fact. There is a number of papers with identification results in this model class (Brito and Pearl, 2002; Tian, 2007, 2009; Foygel et al., 2012; Chen, 2016; Drton and Weihs, 2016; Weihs et al., 2018), though some results only apply to *acyclic* linear SEMs. Bollen (1989) provides an introduction to the model class. We give a short introduction here to describe the connection to Hawkes processes. Let $X = (X_1, \ldots, X_n)^T$ be a vector of random variables and $\varepsilon = (\varepsilon_1, \ldots, \varepsilon_n)^T$ be a vector of zero-mean noise terms. Let

$$X = BX + \varepsilon \tag{3}$$

such that the matrix $B$ has zeros on the diagonal. We let $\Omega_\varepsilon$ denote the covariance of $\varepsilon$. The matrix $I_n - B$ is invertible if and only if $B$ does not have 1 as an eigenvalue and in this case the covariance of $X$, denoted by $C$, is

$$C = (I_n - B)^{-1}\Omega_\varepsilon(I_n - B)^{-T}. \tag{4}$$

Often one will assume that some entries of $B$ and $\Omega_\varepsilon$ are zero which can be encoded by a DMG (also known as a *mixed graph*) such that if $\alpha \to \beta$ is not in the DMG, then $B_{\beta\alpha} = 0$, and if $\alpha \leftrightarrow \beta$ ($\alpha \neq \beta$) is not in the DMG, then $(\Omega_\varepsilon)_{\beta\alpha} = (\Omega_\varepsilon)_{\alpha\beta} = 0$. The error terms are often assumed to be Gaussian. We note that acyclicity is sometimes assumed, i.e., that the DMG corresponding to the model is acyclic, though we are not making this assumption.

We should note that in the linear SEM literature, authors often parametrize the model in Equation (3) using a transposition of $B$, $X = B^T X + \varepsilon$. We use this alternative notation as it aligns better with common notation in the Hawkes process literature.

One immediately sees the similarity between Equations (2) and (4) in that the observable left-hand sides of the equations impose similar restrictions on the model parameters, and we use this similarity to translate results to the Hawkes models. There are two questions that should be addressed. First, the diagonal of $B$ in the linear SEM is often assumed to be zero, and we will handle this by re-writing Equation (2) slightly (Section 3.2). Second, $\Lambda$ is a diagonal matrix in Equation (2) whereas no such restriction on $\Omega_\varepsilon$ is necessary. We will see that this discrepancy vanishes when we also consider partially observed Hawkes processes (Section 3.1).

### 3.1. Marginalization

If the Hawkes process is partially observed, only a proper submatrix of $\Sigma$ in Equation (2) is available, and the coordinate processes, $V$, are partitioned into a set of observed processes, $O$, and a set of unobserved processes, $U$, such that $V = O \,\dot\cup\, U$. We use a *latent projection* to represent the graphical structure of the partially observed system. We say that edges $\alpha \to \beta$ and $\alpha \leftrightarrow \beta$ have a *head* at $\beta$.

**Definition 3 (Latent projection, Verma and Pearl (1990); Richardson et al. (2017))**
*Let $\mathcal{G} = (V, E)$ be a DMG. Its latent projection on nodes $O$ is the DMG $\bar{\mathcal{G}} = (O, F)$ such that for $\alpha, \beta \in O$ we have $\alpha \to \beta$ in $\bar{\mathcal{G}}$ if there is a directed walk from $\alpha$ to $\beta$ such that every non-endpoint node is in $V \setminus O$. We have $\alpha \leftrightarrow \beta$ in $\bar{\mathcal{G}}$ if there is a walk between $\alpha$ and $\beta$ with heads at both $\alpha$ and $\beta$ such that every non-endpoint node is a noncollider and is in $V \setminus O$.*

Note that for a walk in the above definition the set of non-endpoint nodes can be empty and therefore if $\alpha \to \beta$ is in $\mathcal{G}$, then it is also in its latent projection whenever $\alpha, \beta \in O$. We also note that the definition uses walks to allow for cycles (Mogensen and Hansen, 2020). In DAG-based models, this is not needed and the definition is often stated using paths (Richardson et al., 2017).

Using a DMG, $\mathcal{G} = (V, E)$, we will describe a parametrization of the observed integrated covariances. We let $\mathbb{M}_+^s(\mathcal{G})$ denote the set of $(n \times n)$ Schur stable matrices with nonnegative entries such that for $\alpha, \beta \in V = \{1, 2, \ldots, n\}$ and $M \in \mathbb{M}_+^s(\mathcal{G})$

$$\alpha \not\to_\mathcal{G} \beta \Rightarrow M_{\beta\alpha} = 0.$$

We let $\mathbb{M}_{+,0}^s(\mathcal{G}) \subseteq \mathbb{M}_+^s(\mathcal{G})$ denote the subset of such matrices in which all diagonal elements are zero. We let $\mathbb{M}_+^{pd}(\mathcal{G})$ denote the set of symmetric, positive definite matrices with nonnegative entries such that for $\alpha, \beta \in V, \alpha \neq \beta$, and $M \in \mathbb{M}_+^{pd}(\mathcal{G})$

$$\alpha \not\leftrightarrow_{\mathcal{G}} \beta \Rightarrow M_{\beta\alpha} = M_{\alpha\beta} = 0.$$

**Theorem 4** *Let $X$ be a multivariate Hawkes process such that $\rho(\Phi) < 1$ and such that $O$ are the observed coordinate processes and $U$ are the unobserved coordinate processes. Let $\mathcal{D} = (O \dot{\cup} U, E)$ be its causal graph. Then the observed integrated covariance can be written as*

$$\Sigma = (I - \Xi)^{-1}\Theta(I - \Xi)^{-T} \tag{5}$$

*such that $(\Xi, \Theta) \in \mathbb{M}_+^s(\mathcal{G}) \times \mathbb{M}_+^{pd}(\mathcal{G})$ when $\mathcal{G}$ is the latent projection of $\mathcal{D}$ on $O$.*

**Proof** Let $(\Phi, \Lambda)$ denote the parameters in the full system, and let $\Sigma$ denote the full integrated covariance matrix. Let $\Sigma_{OO}$ be the observed submatrix of $\Sigma$, $k = |O|$, and $l = |U|$. We have that $I - \Xi$ is invertible and

$$\Sigma_{OO} = (I - \Xi)^{-1}\Theta(I - \Xi)^{-T} \tag{6}$$

where

$$\Xi = \Phi_{OO} + \Phi_{OU}(I_l - \Phi_{UU})^{-1}\Phi_{UO}$$
$$\Theta = (I_k - \Xi)[(I_n - \Phi)^{-1}\Lambda(I_n - \Phi)^{-T}]_{OO}(I_k - \Xi)^T,$$

see Hyttinen et al. (2012). Using Schur complements, we can write (see Appendix A)

$$\Theta = \Lambda_{OO} + \Phi_{OU}((I_n - \Phi)_{UU})^{-1}\Lambda_{UU}((I_n - \Phi)_{UU})^{-T}(\Phi_{OU})^T.$$

$\Theta$ is clearly positive semidefinite, and positive definite if $\Lambda$ is. Then entries of $\Xi$ and $\Theta$ are nonnegative, and Lemma 5 shows that $\Xi$ is Schur stable. If $\alpha$ is not a parent of $\beta$ in the latent projection then there is no directed walk from $\alpha$ to $\beta$ in $\mathcal{D}$ such that every nonendpoint node is unobserved and it follows that $\Xi_{\beta\alpha} = 0$ using that $(I - A)^{-1} = \sum_{i=0}^{\infty} A^i$ when $\rho(A) < 1$. Similarly, we see that $\alpha \not\leftrightarrow_{\mathcal{G}} \beta$ implies that $\Theta_{\alpha\beta} = \Theta_{\beta\alpha} = 0$ for $\alpha \neq \beta$. ∎

**Stability under Marginalization** We use the following lemma to show that if the full system is Schur stable, then a partially observed system is also Schur stable. This result does not hold without the assumption of nonnegative entries in the directed matrix (see Hyttinen et al. (2012) for an example that illustrates this).

**Lemma 5** *Let $A$ be a square matrix with nonnegative entries such that $\rho(A) < 1$, and let $O \dot\cup U$ be a partition of its rows and columns. The matrix $B = A_{OO} + A_{OU}(I - A_{UU})^{-1}A_{UO}$ has nonnegative entries and $\rho(B) < 1$.*

**Proof** The matrix $A_{UU}$ is a principal submatrix of $A$ and therefore $\rho(A_{UU}) \leq \rho(A) < 1$ (Horn and Johnson, 1985, Corollary 8.1.20). Therefore $(I - A_{UU})^{-1} = \sum_{i=0}^{\infty}(A_{UU})^i$ and it follows that every entry of $B$ is nonnegative.

It holds that $I - B$ is the Schur complement of $I - A_{UU}$ in $I - A$, that is, $I - B = (I - A)/(I - A_{UU})$, and therefore $I - B$ is invertible. The matrix $I - A$ is a nonsingular $M$-matrix so $(I - A)^{-1}$ has nonnegative entries. It follows that $(I - B)^{-1}$ has nonnegative entries and therefore $I - B$ is a nonsingular M-matrix (Berman and Plemmons, 1979, Chapter 6, Theorem 2.3). Then every real eigenvalue of $I - B$ is positive (Berman and Plemmons, 1979, Chapter 6, Theorem 2.3). Since $B$ is nonnegative, $\rho(B)$ is an eigenvalue of $B$ (Horn and Johnson, 1985, Theorem 8.3.1). It follows that $1 - \rho(B)$ is an eigenvalue of $I - B$ and therefore $1 - \rho(B) > 0$. ∎

### 3.2. Normalized Effects

When $\Phi$ is a matrix of direct effects of a Hawkes process (or marginalized effects, $\Xi$), the diagonal elements are not necessarily zero. On the other hand, in a linear SEM the diagonal of $B$ (Equation (3)) is commonly assumed to be zero. We introduce the matrix $\Gamma$ by scaling each row of $\Xi$ by one minus its diagonal element and placing zeros on the diagonal,

$$\Gamma_{\beta\alpha} = \Xi_{\beta\alpha}/(1 - \Xi_{\beta\beta}) \text{ if } \beta \neq \alpha, \text{ and } \Gamma_{\beta\alpha} = 0 \text{ if } \beta = \alpha,$$

and $I - \Gamma$ is invertible (Hyttinen et al., 2012). Hyttinen et al. (2012) study linear SEMs and use the same approach to derive so-called canonical linear cyclic models from models with self-loops. One should note that $\Xi_{\beta\beta} \leq \rho(\Xi)$ (Horn and Johnson, 1985, Corollary 8.1.20). Therefore, if $\Xi$ corresponds to a partially observed linear Hawkes process with $\rho(\Phi) < 1$, we have $\rho(\Xi) < 1$ (Lemma 5) and $\Xi_{\beta\beta} \neq 1$.

We think of the above operation as a type of normalization, and if this is done in the full model (no marginalization), we will say that the normalized parameters, that is, $\Phi_{\beta\alpha}/(1 - \Phi_{\beta\beta})$, are *normalized direct (causal) effects*. It is seen that a normalized direct effect, $\Phi_{\beta\alpha}/(1 - \Phi_{\beta\beta}), \alpha \neq \beta$, is the average number of direct descendants of type $\beta$ from an $\alpha$-event, counting every direct self-triggered event in the $\beta$-process. That is, on the Hawkes tree rooted at $\alpha$, $\Phi_{\beta\alpha}/(1 - \Phi_{\beta\beta})$ is the expected number of $\beta$-events on subtrees of the form $\alpha - \beta - \beta - \ldots - \beta$ for any number of $\beta$-events.

**Corollary 6** *Let $X$ be a multivariate Hawkes process with stationary increments such that $O$ are the observed coordinate processes and $U$ are the unobserved coordinate processes and let $\mathcal{D}$ be its causal graph. Then the observed integrated covariance can be written as*

$$\Sigma = (I - \Gamma)^{-1}\Omega(I - \Gamma)^{-T}$$

*such that $(\Gamma, \Omega) \in \mathbb{M}_{0,+}^s(\mathcal{G}) \times \mathbb{M}_+^{pd}(\mathcal{G})$ when $\mathcal{G}$ is the latent projection of $\mathcal{D}$ on $O$.*

**Proof** Consider the representation from Theorem 4. Then we have

$$\Sigma = (I - \Phi)^{-1}\Theta(I - \Phi)^{-T}.$$

Let $D$ denote a diagonal matrix such that the $\alpha$'th diagonal element equals $1 - \Phi_{\alpha\alpha}$ and $\Gamma$ is as defined above. Then

$$\begin{aligned}
\Sigma &= (D(I - \Gamma))^{-1}\Theta(D(I - \Gamma))^{-T} \\
&= (I - \Gamma)^{-1}D^{-1}\Theta D^{-1}(I - \Gamma)^{-T}.
\end{aligned} \tag{7}$$

The claims all follow from Theorem 4. We should just show that $\Gamma$ is a Schur stable matrix (Proposition 7). ∎

If we consider a Hawkes process and its causal graph, $\mathcal{D} = (V, E)$, we see that the latent projection of $\mathcal{D}$, $\mathcal{G} = (O, F)$, $O \subseteq V$, represents the observed covariance, $\bar{\Sigma} = \Sigma_{OO}$ in the following way. If $\alpha \to \beta$ is not in $\mathcal{G}$, then $\Gamma_{\beta\alpha} = 0$. For $\alpha \neq \beta$, if $\alpha \leftrightarrow \beta$ is not in $\mathcal{G}$, then $\Omega_{\beta\alpha} = \Omega_{\alpha\beta} = 0$. Omitting self-loops, this corresponds to the graphical representation of a linear SEM. This means that the observable integrated covariances after rewriting are parametrized exactly like a linear SEM and we will use this to obtain identification results (Section 4) and equality constraints (Section 5).

While the above corollary casts the Hawkes integrated covariance as a set of equations describing the covariance matrix of a linear SEM, the Hawkes parameters are required to be nonnegative.

**Proposition 7** *Let $\Xi$ be a nonnegative matrix such that $\rho(\Xi) < 1$. Then the normalized matrix, $\Gamma$, also satisfies $\rho(\Gamma) < 1$.*

## 4. Identification

Using constraints on the causal graph, we will discuss the identification of the entries of the matrices $\Xi$ and $\Gamma$. To give a formal definition of identification, consider two stationary Hawkes processes, $M_1 \in \mathbb{M}$ and $M_2 \in \mathbb{M}$, with coordinate processes $V_1$ and $V_2$, respectively, for a model $\mathbb{M}$. Let $\tilde{P}_1$ and $\tilde{P}_2$ be their marginal distributions over a common observed set of coordinate processes, $O$, let $f$ be a function of such a marginal distribution, and let $\alpha, \beta \in O$. Finally, let $\Pi_{\beta\alpha}$ be a parameter ($\Pi_{\beta\alpha} = \Xi_{\beta\alpha}$ or $\Pi_{\beta\alpha} = \Gamma_{\beta\alpha}$). We will say that the parameter $\Pi_{\beta\alpha}$ *is identified from* $f(\tilde{P})$ in the model $\mathbb{M}$ if $f(\tilde{P}_1) = f(\tilde{P}_2)$ implies $\Pi_{\beta\alpha}^1 = \Pi_{\beta\alpha}^2$ where $\Pi_{\beta\alpha}^i$ is the parameter value for $M_i$. Note that while identification is sometimes only discussed from the observed distribution in its entirety, we use $f$ to distinguish simpler statistics that can be sufficient for identification. As we will see, under graphical constraints some parameters are identified from the integrated covariance matrix in which case the inferential problem is reduced to a finite-dimensional one.

### 4.1. Connection to Linear SEMs

There are several papers describing graphical conditions to ensure identifiability of parameters of a linear structural equation model. For these results, one assumes a certain graphical structure represented by a DMG which encodes sparsity in the matrices $B$ and $\Omega_\varepsilon$ in Equation (4). Identification of a parameter means that for a fixed graphical structure and an observed covariance, $C$, this parameter is uniquely determined. The idea is now that one can use any available identification result from the (cyclic) linear structural equation toolbox to provide identification results for the entries in $\Gamma$ such as those provided by, e.g., Foygel et al. (2012); Chen (2016); Weihs et al. (2018). This follows

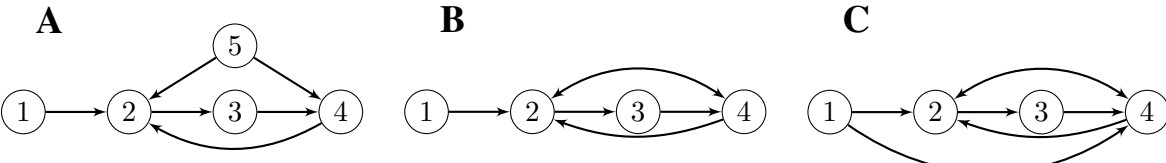

Figure 3: **A**: Directed graph such that each node represents a coordinate process and the edge $\alpha \to \beta$ is present if and only if $\Phi_{\beta\alpha} \neq 0$. **B**: Latent projection of the graph **A** on nodes $O = \{1, 2, 3, 4\}$. **C**: Graphs **B** and **C** are Markov equivalent (entail the same $\mu$-separations), yet are distinguishable using an equality constraint on the observable matrix $\bar{\bar{\Sigma}} = \Sigma_{OO}$. Loops (self-edges) are omitted from graphs **A**, **B**, and **C**.

from noting that if we use the latent projection of the Hawkes causal graph to encode sparsity, the problem in Corollary 6 is the same as the linear SEM problem, though with a smaller parameter space as $\Gamma$ and $\Omega$ are required to be nonnegative and $\Gamma$ is required to be Schur stable. While any identification result from the linear SEM problem therefore applies to the problem in Corollary 6, one can actually obtain stronger results using that the parameter space is smaller. Section 4.2 shows this by an example. We also note that one typically has to distinguish between *global* and *generic* identification of parameters.

**Example 8** *Consider the graph in Figure 3A and assume that this is the causal graph of a Hawkes process on nodes $V = \{1, 2, 3, 4, 5\}$ such that coordinate process 5 is unobserved. Let $\Gamma$ be the $4 \times 4$ matrix of direct effects. If we apply the identification criterion of Weihs et al. (2018), we find that every $\Gamma_{\beta\alpha}$ such that $\alpha, \beta \in O$ is identified from $\Sigma$ (note that $\Gamma_{\beta\alpha} = \Phi_{\beta\alpha}/(1 - \Phi_{\beta\beta})$ for $\alpha, \beta \in O$, $\alpha \neq \beta$, in this example as $\Phi_{UO} = 0$). It follows that $\Phi_{11}, \Phi_{32}$, and $\Phi_{33}$ are identified from the integrated mean and covariance.*

*On the other hand, one can show by direct computation that $\Phi_{21}, \Phi_{22}, \Phi_{24}, \Phi_{43}$, and $\Phi_{44}$ are not identified. If we consider $\Phi_{21}$, we can understand the lack of identication from the fact that $\Phi_{22}$ cannot be identified. Intuitively, this is explained by its unobserved parent process which implies that we cannot, using only the integrated covariance, separate the contributions from coordinate processes 2 and 5 to the observed integrated covariance.*

### 4.2. Identification of Cycles

The $n$-cycle, $\mathcal{D}_n$, is the directed graph on nodes $\alpha_1, \ldots, \alpha_n$ such that $\alpha_i \to \alpha_{i+1}$ for all $i = 1, \ldots, n-1$ and $\alpha_n \to \alpha_1$. When $n \geq 3$, Drton et al. (2011) show that Equation (4) has at most two solutions when the graph is an $n$-cycle and that generically it has two solutions. We show in the following that at most one of them is in $\mathbb{M}_{+,0}^s(\mathcal{G}) \times \mathbb{M}_+^{pd}(\mathcal{G})$, thus, is a solution to the normalized Hawkes problem.

**Proposition 9** *Let $\mathcal{G}$ be an $n$-cycle, $n \geq 3$. There is at most one solution to*

$$\Sigma = (I - \Gamma)^{-1}\Omega(I - \Gamma)^{-T}$$

*such that $(\Gamma, \Omega) \in \mathbb{M}_{0,+}^s(\mathcal{G}) \times \mathbb{M}_+^{pd}(\mathcal{G})$.*

## 5. Equality Constraints

It is well-known that partially observed DAG-models satisfy constraints that are not described by conditional independence some of which are known as *equality constraints*. Richardson et al. (2017) treat the general (acyclic) case while Chen et al. (2014); Chen (2016) consider equality constraints in linear SEMs. In DAG-based models, equality constraints are testable implications that may bring strictly more information about the underlying structure than using conditional independence alone. We will see that the analogous statement holds for the Hawkes process models, though the relevant notion of independence to compare with is that of *local independence*. Local independence is an asymmetric notion of independence of the coordinate processes of a multivariate stochastic process (Schweder, 1970; Aalen, 1987; Didelez, 2000, 2008). The following definition is specific to point processes, however, it can be extended to other classes of stochastic processes (Aalen, 1987; Didelez, 2006; Mogensen et al., 2018; Mogensen and Hansen, 2020).

**Definition 10 (Local independence)** *Let $X = (X^1, \ldots, X^n)^T$ be a multivariate point process with intensity processes $\lambda$ and let $V = \{1, \ldots, n\}$. Let $A, B, C, D \subseteq V$. Let $\mathcal{F}_t^D$ denote the natural filtration of $X^D$, i.e., the filtration generated by the coordinate processes in $D$. We say that $X^B$ is locally independent of $X^A$ given $X^C$ if for all $\beta \in B$ and for all $t \in \mathbb{R}$,*

$$E(\lambda_t^\beta \mid \mathcal{F}_t^{A \cup C})$$

*is adapted to the filtration $\mathcal{F}_t^C$.*

### 5.1. Structure learning in Hawkes processes

A linear Hawkes process has a very simple 'mechanistic' interpretation which is evident from both the intensity representation in Definition 1 and the cluster representation (e.g., Figure 2). In the intensity process of $\beta$, a function $\phi_{\beta\alpha}$ concisely describes how an $\alpha$-event increases the intensity with which future $\beta$-events occur. This makes Hawkes processes attractive in settings where an analyst is not merely trying to fit a process to an observed distribution but is also interested in understanding how events in some coordinate processes create events in other coordinate processes. Examples include social media networks, systems of interacting neurons, and alarm networks in complex engineered systems. In many applications, the observed data only include events in a proper subset of the coordinate processes of the full system.

Local independence has been used for structure learning in stochastic process models (Meek, 2014; Mogensen et al., 2018; Thams, 2019; Mogensen, 2020), analogously to how conditional independence is used for constraint-based structure learning in classical models (Spirtes et al., 2000; Spirtes and Zhang, 2019). Structure learning based on local independence typically assumes *faithfulness*, that is, that $X^B$ is locally independent of $X^A$ given $X^C$ if and only if $B$ is $\mu$-separated from $A$ given $C$ in $\mathcal{G}$, the DMG representing the model ($\mu$-separation, or $\delta$-separation, is a concept analogous to $m$-, or $d$-separation (Didelez, 2000, 2008; Mogensen and Hansen, 2020)). The *Markov equivalence class* of $\mathcal{G}$ is the set of DMGs that encode exactly the same set of $\mu$-separations, hence local independencies. Mogensen and Hansen (2020) show that each Markov equivalence class contains a unique maximal element and this graph is therefore a natural learning target when using tests of local independence.

An obvious question is therefore whether there are constraints imposed by the structure that are not described by local independence, or $\mu$-separation. This is indeed the case and such constraints

can be used to distinguish between graphical structures that imply the same set of local independencies. In the next subsection, we show that one can use the integrated mean and covariance to find such constraints. In applications, this means that the set of graphical structures that can explain the observed data may be reduced, thus we may obtain strictly more information about how coordinate processes interact.

## 5.2. Constraints on the Integrated Covariance

From Equation (7) and the section on marginalization we see that every equality constraint in a linear structural equation model implied by the graphical structure imposes the same constraint on the parameters $(\Gamma, \Omega)$ and this allows us to find constraints in the Hawkes model. Chen et al. (2014) describe how to find equality constraints in linear SEMs, more specifically through *overidentification* of model parameters. When parameters in the linear SEM are identified they can be computed from the observed covariance matrix. Overidentification occurs when there exist two ways to identify a parameter through different functions of the covariance matrix and based on logically independent model assumptions. An equality constraint arises from equating such functions. Chen et al. (2014) provide algorithms for systematically finding different identifying functions based on different parts of the model structure.

We will show by an example that equality constraints may provide information about the underlying structure of a linear Hawkes process which is not contained in its ($\mu$-separation) Markov equivalence class.

**Example 11** *Consider again the example illustrated by the graph in Figure 3B. If we think of the graph as representing a linear SEM, we can, e.g., use Theorem 1 and Lemma 1 of Chen et al. (2014) to see that the graphical structure imposes the constraint*

$$\Sigma_{14}/\Sigma_{13} = (\Sigma_{34} - \Gamma_{32}\Sigma_{24})/(\Sigma_{33} - \Gamma_{32}\Sigma_{23}),$$

*and therefore this algebraic constraint is also satisfied by the Hawkes model corresponding to the graph (note that $\Gamma_{32}$ is identified and that the constraint corresponds to two ways of identifying $\Gamma_{43}$). The graphs **B** and **C** in Figure 3 (right) imply the same set of local independencies when using $\mu$-separation. However, the above constraint is satisfied in **B** only and therefore this constraint allows us to discriminate between the two models.*

In DAG-based models, equality constraints may be thought of as conditional independence constraints in identified interventional distributions (Richardson et al., 2017). This is also seen to be the case in the linear Hawkes setting. First, we define a type of intervention different from the injection intervention in Section 2.1. Assume we have a linear Hawkes process as defined by a set of link functions $\{\phi_{\beta\alpha}\}_{\alpha,\beta\in V}$ and a set of constants $\{\mu_\alpha\}_{\alpha\in V}$ with (normalized) parameters $(\Gamma, \Omega)$ over the observed set of coordinate processes, $O$. In this intervention and for a subset of coordinate processes $I \subseteq O \subseteq V$, we set $\bar{\phi}_{\beta\alpha} = 0$ and $\bar{\mu}_\beta = 1$ whenever $\beta \in I$ and otherwise $\bar{\phi}_{\beta\alpha} = \phi_{\beta\alpha}$ and $\bar{\mu}_\beta = \mu_\beta$. The intervened process is the linear Hawkes process defined by $\{\bar{\phi}_{\beta\alpha}\}_{\alpha,\beta\in V}$ and $\{\bar{\mu}_\alpha\}_{\alpha\in V}$. Note that $\bar{\Phi}$ is Schur stable. This intervention implies that all events in the processes $I$ are exogenous (generation 0 in the cluster representation). Consider also the corresponding linear SEM with variables indexed by $O$ such that $B = \Gamma$ and $\Omega_\varepsilon = \Omega$ in Equation (4). The observed integrated Hawkes covariance in the interventional process is then equal to the interventional covariance in this linear SEM when variables in $I$ are made exogenous with zero mean and unit variance. This means

that when equality constraints can be thought of as constraints in an interventional distribution in the linear SEM, they can also be thought of as constraints on the observed integrated Hawkes covariance of an interventional process.

## 6. Conclusion

We obtained identification results and equality constraints for Hawkes process models through the similarity between the integrated Hawkes covariance and the observed covariance of a linear structural equation model. These equality constraints are useful for structure learning as they can supplement constraint-based learning based on tests of local independence. The approach in this paper finds equality constraints through a time-independent statistic which may be applicable in other classes of stochastic processes as well.

A Hawkes process is defined using the set of $\phi_{\beta\alpha}$-functions along with the $\mu_\alpha$-constants for $\alpha, \beta$ in the finite set $V$ and a Hawkes process model is in that sense nonparametric. Constraints that arise from the integrated covariance offer also a dimension reduction as these constraints exist in a finite-dimensional parameter space instead of a function space. This makes them more suitable to employ in data analysis as estimation of the link functions, $\{\phi_{\beta\alpha}\}$, is a challenging problem even in the case of full observation and partial observation only adds to the complexity of this task.

## Acknowledgments

This work was supported by a research grant from VILLUM FONDEN (13358) and by a DFF-International Postdoctoral Grant (0164-00023B) from Independent Research Fund Denmark. The author thanks Niels Richard Hansen, Ilya Shpitser, and Daniel Malinsky for helpful discussions. The author is also grateful to the reviewers for their comments and suggestions.

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

## Appendix A. Marginalization

In this section and for a matrix $A$, we define $A_{\bar{I}J}^{-1} = (A^{-1})_{\bar{I}\bar{J}}$ and $A_{\bar{I}J}^{-T} = (A^{-T})_{\bar{I}\bar{J}}$. Identity matrices of different dimensions are all denoted by $I$. Consider an invertible block matrix

$$M = \begin{pmatrix} A & B \\ C & D \end{pmatrix}$$

such that $D$ is invertible. We let $M/D = A - BD^{-1}C$ denote the *Schur complement* of the block matrix $D$ in the matrix $M$. It holds that

$$M^{-1} = \begin{pmatrix} (M/D)^{-1} & -(M/D)^{-1}BD^{-1} \\ -D^{-1}C(M/D)^{-1} & D^{-1}(I + C(M/D)^{-1}BD^{-1}) \end{pmatrix}.$$

Matrices $I - \Phi$ and $I - \Phi_{UU}$ are both invertible as $I - \Phi$ is a nonsingular M-matrix (Chapter 6, Theorem 2.3 in Berman and Plemmons (1979)). Therefore,

$$(I - \Phi)_{OO}^{-1} = ((I - \Phi)/(I - \Phi)_{UU})^{-1} = (I - \Xi)^{-1}$$

by definition of $\Xi$. Furthermore,

$$(I - \Phi)_{OU}^{-1} = -(I - \Xi)^{-1}(I - \Phi)_{OU}((I - \Phi)_{UU})^{-1}$$

Let $I = I_n$ and $A_{\bar{I}} = A_{\bar{I}\bar{I}}$. From the above, it follows that

$$\begin{aligned}
\Theta &= (I_k - \Xi)[(I_n - \Phi)_O^{-1}\Lambda_O(I - \Phi)_O^{-T} + (I - \Phi)_{OU}^{-1}\Lambda_U(I - \Phi)_{UO}^{-T}](I_k - \Xi)^T \\
&= \Lambda_O + (I_k - \Xi)(I - \Phi)_{OU}^{-1}\Lambda_U((I - \Phi)_{UO}^{-T}(I_k - \Xi)^T \\
&= \Lambda_O + (I - \Phi)_{OU}((I - \Phi)_U)^{-1}\Lambda_U((I - \Phi)_U)^{-T}((I - \Phi)_{OU})^T \\
&= \Lambda_O + \Phi_{OU}((I - \Phi)_U)^{-1}\Lambda_U((I - \Phi)_U)^{-T}(\Phi_{OU})^T.
\end{aligned}$$

## Appendix B. Proofs

**Proof** [Proposition 7] Assume to obtain a contradiction that $\rho(\Gamma) \geq 1$ and let $\lambda = \rho(\Gamma)$. The matrix $\Gamma$ is nonnegative and therefore $\lambda$ is an eigenvalue of $\Gamma$ and a nonnegative eigenvector, $x \neq 0$, can be chosen (Horn and Johnson, 1985, Theorem 8.3.1), $\Gamma x = \lambda x$.

Let $D_\Xi$ denote the diagonal matrix such that $(D_\Xi)_{ii} = \Xi_{ii}$ for all $i$. From the definition of $\Gamma$, we have

$$\Xi = \Gamma - D_\Xi\Gamma + D_\Xi.$$

Multiplying with $x$ and using $\Gamma x = \lambda x$, we obtain

$$\Xi x = \lambda x + (1 - \lambda) D_\Xi x$$

For all $i$, $(D_\Xi)_{ii} = \Xi_{ii} \le \rho(\Xi) < 1$ (Horn and Johnson, 1985, Corollary 8.1.20). We have that $1 - \lambda \le 0$ and that the entries of $x$ are nonnegative and therefore

$$\Xi x \ge \lambda x + (1 - \lambda) x = x$$

where the inequality should be read entrywise. It holds that $x \ne 0$ and therefore $\Xi x \ge x$ implies that $\rho(\Xi) \ge 1$ which is a contradiction (Horn and Johnson, 1985, Theorem 8.3.2). ∎

**Proof** [Proposition 9] Let $(\Gamma_0, \Omega_0)$ ,$(\Gamma_1, \Omega_1)$ be the two solutions of a linear SEM corresponding to an $n$-cycle (generically, they are distinct). Let $\delta_i^j$ denote the inverse of the $i$'th diagonal element of $\Omega_j$ and let $\lambda_i^j$ denote the $(i, i+1)$-entry of $\Gamma_j$ (modulo $n$). Drton et al. (2011) show that if $\lambda_i^0 = 0$ for some $i$ then there is at most one solution. Assuming that $\lambda_i^0 \ne 0$ for all $i = 1, \ldots, n$, they show that

$$\delta_i^1 = \delta_i^0 + \left( \prod_j \delta_j^0 \right) \left( \left[ \prod_j \lambda_j^0 \right]^2 - 1 \right) / \det(K_{-i}) \qquad \text{and}$$

$$\delta_i^0 = \delta_i^1 + \left( \prod_j \delta_j^1 \right) \left( \left[ \prod_j \lambda_j^1 \right]^2 - 1 \right) / \det(K_{-i}),$$

where $K$ is the inverse of the observed covariance, $\Sigma$, and $K_{-i}$ denotes the submatrix of $K$ with the $i$'th row and column deleted. Replacing the first $\delta_i^1$ in the lower equation with the expression in the upper equation we obtain,

$$- \left( \prod_i \delta_i^1 \right) \left( \left[ \prod_i \lambda_i^1 \right]^2 - 1 \right) = \left( \prod_i \delta_i^0 \right) \left( \left[ \prod_i \lambda_i^0 \right]^2 - 1 \right). \tag{8}$$

We assume that the observed covariance is from a Schur stable system, say $\rho(\Gamma_0) < 1$. Then every eigenvalue of $\Gamma_0$ has absolute value less than 1, and $|\det(\Gamma_0)| < 1$. Due to the structure of $\Gamma_0$, $|\det(\Gamma_0)| = \prod_i \lambda_i^0$ such that $\prod_i \lambda_i^0 < 1$ and $\prod_i (\lambda_i^0)^2 < 1$. Every $\delta_i^0$ is positive, and this means that the right-hand side of Equation (8) is negative. Every $\delta_i^1$ and every $\lambda_i^0$ are positive and therefore we must have $(\prod_i \lambda_i^1)^2 - 1 > 0$ and $\prod_i \lambda_i^1 > 1$. This means that $|\det(\Gamma_1)| > 1$ and therefore $\rho(\Gamma_1) \ge 1$. ∎

