# OpenReview forum: "Equality Constraints in Linear Hawkes Processes"
_cclear.cc/CLeaR/2022/Conference — CLeaR 2022 Poster_

### Official Review · Reviewer_P9hg · 2021-11-22

**Confidence:** 3
**Overall Score:** 7

**Main Review:**

Identification of a model is an important topic, especially in causal inference.\
The paper builds on prior work in the field of Hawkes processes and testable constraints (latent projection and algebraic equality constraints). \
The authors study the new identification results for a well-known Hawkes process when there exist some unobserved variables. \
Though most of technics have been proposed, their conclusions in Hawkes processes appear both novel and interesting.

It would be helpful to describe some real-world implications of their conclusions.

I might have some misunderstandings about this paper. So it is very likely that I will change my score if the other reviewers provide some new points.

**Summary:**

This paper considers the problem of identification of the linear Hawkes Processes . The main contributions is that the authors transfer the linear Hawkes processes to a linear SEM under some assumptions and provide the identification results of linear Hawkes processes by using the existing results of linear SEM.

---

> ### Author Response · Authors · 2021-12-03
> **Reply**
>
> Thank you for your comments. We'll add some description of real-world applications. We believe that in many applications of Hawkes processes, analysts only have partial observation (some coordinate processes are unobserved) and due to the nature of the Hawkes process (very explicit interactions modeled by what we call causal link functions), it is often used when analysts are interested in an underlying 'structure' or causal network. Therefore, structure learning in Hawkes processes finds application in the study of, e.g., social networks and interacting neurons.

---

> > ### Comment · Reviewer_P9hg · 2021-12-11
> > **Thanks**
> >
> > Thank you for your clarification. I think this is a good paper, and my score remains unchanged.

---

### Official Review · Reviewer_id5m · 2021-11-22

**Confidence:** 3
**Overall Score:** 7

**Main Review:**

Originality:

The originality of this work is primarily in noting that graphical models of
Hawkes processes are closely related to linear structural equations models, and
in demonstrating properties of the parameters of the model. This is an original
insight, and might motivate discovery of additional properties of such models.

Significance:

The theory of causal inference in continuous time settings is not well
developed. This work makes some progress on developing this theory, and will
likely be of use to analysts working in these settings.

That said, the main _causal_ results of this paper in Sections 4 and 5 have a somewhat sparse presentation. The approach taken to presenting the results of these sections is that once the
relationship between SEMs and Hawkes process models is established, results from
linear SEMs can be used in the Hawkes process models under consideration. A
simple example of the titular equality constraints is presented illustrating a
setting in which the novel equality constraints can help adjudicate between
candidate models (Figure 3 B, C), but it is not clear when these constraints
arise in general. It would be helpful to see some discussion of graphical
conditions under which these constraints arise, and when they are non-trivial,
even if that takes some recounting of material from Chen et al 2014. Such
elaborations may be added as supplementary materials, as they will not
necessarily be original contributions.

Technical Quality and Clarity:

The paper is clearly written and seems technically correct, though I do not have
the necessary background to validate all the proofs.



**Summary:**

Summary:  The authors prove the existence of equality constraints in graphical models of Hawkes processes, and posit that these constraints can be used for structure-learning tasks.

---

> ### Author Response · Authors · 2021-12-03
> **Reply**
>
> Thank you for your comments. We would be happy to recount some of the results from Chen et al (2014) to make our paper more self-contained, in particular describing how to find equality constraints in a more general fashion. Adding a description of the connection between zero-interventions in Hawkes processes and interventions in linear SEMs (see 1 in the reply to Reviewer zgzK) may also help explain the causal interpretation better.

---

### Official Review · Reviewer_zgzK · 2021-11-23

**Confidence:** 5
**Overall Score:** 7

**Main Review:**

Strengths:

- The results are presented clearly and intuitively: the connection and explanations provided via ordinary linear SEMs is much appreciated.
- Similar to Verma constraints for ADMGs, the results on equality constraints here may be translated into improvements in structure learning algorithms. That is, more informative structures can be learned by algorithms that take advantage of such constraints.

Weaknesses:

- I think some of the theory could have been pushed a bit further -- many of the results stem from applications of existing criteria phrased for ADMGs.

Open questions for the authors:

- Are there ways to think of these generalized equality constraints in linear Hawkes processes as \mu-separation in a conditional DMG, just as we are able to some Verma constraints as m-separation in a conditional ADMG. That is, can these be phrased as constraints that appear in post-interventional distributions? At first glance it would seem so -- \mu-separation between 1 and 4 should hold after cutting incoming edges (intervening) on 3? I wonder if that would lead to an interesting theory in the non-parametric form of these constraints.

- Is there always an equality constraint implied by a missing edge? I think this probably isn't the case, because it's not true for regular ADMGs. Sorry for answering my own question there, but hopefully the follow up is also interesting -- is there a notion of maximality that can be defined here, similar to MAGs for ordinary conditional independence and MArGs for generalized equality constraints? The latter might be quite hard, but it may be worth thinking of a maximality condition for ordinary constraints (if there isn't one already)? Because maximal graphs form the natural units for structure learning in some sense.

Additional related works:

The paper is very thorough in its literature review. I only have a few suggestions that I think are relevant to the introductory discussion on structure learning algorithms that take advantage of generalized equality (Verma) constraints. Many of the references listed there currently are technically papers that lay out theory for what these equality constraints may look like without proposing specific algorithms to take advantage of them.
- Structure learning with equality constraints in linear Gaussian SEMs -- https://arxiv.org/pdf/2010.06978.pdf
- Structure learning with equality constraints in linear non-Gaussian SEMs -- https://arxiv.org/pdf/2007.11131.pdf

**Summary:**

The paper presents causal identification theory and testable implications for linear Hawkes' processes with unmeasured confounders. The results are novel, interesting, and present interesting paths forward for possible extensions.

---

> ### Author Response · Authors · 2021-12-03
> **Reply**
>
> Thank you for your comments and questions. These are partial answers to the open questions:
>
> 1) There is a way to think about some of these constraints as induced by m-separation in conditional (A)DMGs. First, define a zero-intervention on process i in the Hawkes process to be an intervention which sets the causal link functions into i to be zero and also sets the baseline intensity (mu_i) to zero (this means that no events occur in process i). Consider also the corresponding linear SEM in which we define an intervention on variable i, X^i, by simply removing the i'th equation and keeping the other equations (setting the random variable X^i equal to some fixed number x^i in the other equations). Then the interventional (under a zero-intervention) Hawkes integrated covariance equals the covariance matrix of the interventional distribution of the linear SEM. This means that whenever there's a constraint on the covariance matrix of the linear SEM which is implied by m-separation in a conditional (A)DMG, this constraint also applies to the Hawkes integrated covariance (the graphs representing the linear SEM and the Hawkes process are the same so we can construct such conditional (A)DMGs from either). One should note that the notion of causality defined in the paper is essentially a type of 'soft intervention' which is different from the zero-intervention above. However, the zero-intervention has this simple connection to the interventional covariances of the linear SEM. The above uses m-separation (not \mu-separation) and perhaps this is natural as the Hawkes integrated covariance is something similar to a covariance matrix, i.e., there's no explicit time. This argument uses the fact that m-separation in conditional (A)DMGs may imply equality constraints. To our knowledge, this is only well understood in ADMGs, and not in the cyclic case.
>
> 2) Yes, there is such a notion in the case of the ordinary conditional independence relation (for these graphs, this is (conditional) local independence, a notion very similar to Granger causality). The paper by Mogensen and Hansen which is cited in the manuscript proves that there's a unique maximal element within each Markov equivalence class (using \mu-separation instead of m-separation) of DMGs. As argued in that paper, this solves the maximality question as learning based on tests of local independence can output that unique graph. The paper also contains a method for visualizing the entire Markov equivalence class starting from the maximal element of the class. For generalized constraints, we agree that it seems difficult and we probably need to first better understand the generalized constraints.
>
> We'll update the paper with these partial answers. Thank you also for the additional references. It's definitely useful to include references to some papers that tackle the actual learning problem using data.

---

> > ### Comment · Reviewer_zgzK · 2021-12-14
> > **Thank you for the response**
> >
> > Thank you for the thoughtful response, and additional insights. I think this is a good paper, and would love to see it accepted

---

### Decision · Program_Chairs · 2022-01-12

**Decision:**

Accept (Poster)

**Comment:**

The authors study causal identifiability for linear Hawkes Processes by casting the processes into a linear SEM under appropriate assumptions and provide the identification results for linear Hawkes processes using the existing results for linear SEM. The reviewers appreciated the contribution as well as the quality of presentation.